# Genome mining shows that retroviruses are pervasively invading vertebrate genomes

Jianhua Wang[1] & Guan-Zhu Han [1]✉

Endogenous retroviruses (ERVs) record past retroviral infections, providing molecular archives for interrogating the evolution of retroviruses and retrovirus-host interaction. However, the vast majority of ERVs are not active anymore due to various disruptive mutations, and ongoing retroviral invasion of vertebrate genomes has been rarely documented. Here we analyze genomics data from 2004 vertebrates for mining invading ERVs (ERVi). We find that at least 412 ERVi elements representing 217 viral operational taxonomic units are invading the genomes of 123 vertebrates, 18 of which have been assessed to be threatened species. Our results reveal an unexpected prevalence of ongoing retroviral invasion in vertebrates and expand the diversity of retroviruses recently circulating in the wild. We characterize the pattern and nature of ERVi in the historical and biogeographical context of their hosts, for instance, the generation of model organisms, sympatric speciation, and domestication. We suspect that these ERVi are relevant to conservation of threatened species, zoonoses in the wild, and emerging infectious diseases in humans.

Retroviruses exclusively infect vertebrates, causing a variety of diseases, such as AIDS and cancers. Unique among RNA viruses, the replication of retroviruses requires reverse transcription and integration into the host genomes. When infecting germline cells, integrated proviruses may become vertically inherited from parents to offspring, forming endogenous retroviruses (ERVs)[1–3]. ERVs are highly abundant in vertebrate genomes; for instance, ERVs account for more than 8% of the human genome. Replicated as a permanent part of the host genome, ERVs provide molecular fossils for interrogating the deep evolution of retroviruses and the evolution of retrovirus-host interaction[1–3].

Upon integration into the host genome, 5′- and 3′-long terminal repeats (LTRs) of an ERV are identical, and then they accumulate mutations independently as distinct genetic loci[1–3]. ERVs undergo intricate evolutionary dynamics at the level of both genome and population. ERVs amplify in the host genomes through three main mechanisms, namely reinfection (requiring self-encoded intact *gag*, *pol*, and *env* genes), retrotransposition in *cis* (requiring self-encoded intact *gag* and *pol* genes), and complementation in *trans* (with the help of co-infecting retroviruses or through recombination between co-packaged defective retroviral genomes)[4–6]. The fate of each newly generated ERV is governed by natural selection and genetic drift, resulting in insertional polymorphism for some ERVs in the early phases of endogenization. Ultimately, ERV insertions get fixed or lost in the host population. Indeed, the vast majority of ERVs in the vertebrate genomes possess disruptive mutations and are not active anymore. Ongoing retroviral invasion into the vertebrate genomes has been rarely documented, and only a few credible examples exist, such as koala retrovirus A (KoRV-A)[7].

In this study, we performed phylogenomic mining of retroviruses that are invading their host genomes (designated ERVi) from genomics data of 2004 vertebrate species and identified 412 ERVi elements in the genomes of 123 vertebrates. Our pan-vertebrate mining reveals an unexpected prevalence of ongoing retroviral invasion in vertebrates and expands our knowledge of the diversity of retroviruses that are likely to be recently circulating in the wild. Understanding of ongoing retroviral invasion might be of significance to the conservation of threatened species, the surveillance of zoonotic diseases in the wild, and the prediction of emerging infectious diseases in humans.

[1]College of Life Sciences, Nanjing Normal University, Nanjing, China. ✉e-mail: guanzhu@njnu.edu.cn

## Results

### Unexpected prevalence of ongoing retroviral invasion in vertebrates

We analyzed genomics data from 2004 vertebrate species to systemically mine retroviruses invading their host genomes. We used the following criteria to attribute an ERV to an ERVi: First, its LTRs are identical; Second, it encodes intact *gag*, *pro*, *pol*, and *env* genes, indicating it is likely to produce functional virions[4,5]; Third, it exhibits insertional polymorphism in its host population (Fig. 1). Together, we identified a total of 412 ERVi elements that are invading the genomes of 123 vertebrate species (Fig. 2a), suggesting that the genomes of at least 6% of vertebrate species are being invaded by retroviruses. These 412 ERVi elements can be assigned to at least 217 distinct viral operational taxonomic units (vOTUs) based on sequence identity (Fig. 2b, c). Elements within a single vOTU are largely derived from amplification accompanying invasion with Mate.ERVi.2a and Mate.ERVi.2b from *Molothrus ater* as an exception, which was derived from recent chromosomal duplication. Our simulation estimates that ~8338 (95% confidence intervals: 8048 to 8643) ERVi vOTUs are invading the genomes of 74,140 currently described vertebrates[8] (Fig. S1). It should be noted that the prevalence of ERVi is still much underestimated, because population genomics data are only available for a limited number of vertebrates and non-reference ERVs are present in the vertebrate genomes[9]. Given that a pair of identical LTRs may have evolved for hundreds of thousands of years[10], we identified orthologous loci across closely related species for each ERVi element but did not identify any orthologous insertion, further supporting the recency of ERVi identified in this study. Phylogenetic analyses (Fig. 2b) show that most of ERVi fall out of the diversity of retroviruses classified by the International Committee on Taxonomy of Viruses (ICTV)[11], and many ERVi cannot be readily classified into certain genera[12]. Taken together, given that ERVi are viruses in transition between exogenous and endogenous status[7], our study expands the diversity of retroviruses that have been recently circulating in the wild.

ERVi are widespread across the major groups of vertebrates, and the seeming absence of ERVi in some vertebrate groups is likely to be mainly due to limited genome-scale data available (Fig. 2d, e). A high proportion (50/123, 40.65%) of vertebrate species were co-invaded by at least two ERVi vOTUs (Fig. 2d, e). In an extreme case, eight ERVi vOTUs are invading the genomes of mice (*Mus musculus*).

### Phylogenetic signal of retroviral invasion in vertebrates

We found that ERVi invasion exhibits moderately strong phylogenetic clumping in vertebrates using *D* statistics, a measure of phylogenetic signal independent of phylogeny size and trait prevalence[13] (Table 1). These results indicate that ERVi invasion is not random across the vertebrate phylogeny. Among the major lineages of vertebrates, we detected significant phylogenetic signals in amphibians and fishes, but not in reptiles, birds, or mammals (Table 1). Therefore, certain biological or life-history traits might drive the occurrence of retroviral invasion in amphibians and fishes.

### Historical and biogeographical pattern of invading retroviruses

We characterized the pattern and nature of individual ERVi elements in the historical and biogeographical context of their hosts (for each ERVi, see Figs. 3–6 and Supplementary Figs. 2–11 for the details) and here highlighted the following three intriguing cases.

In the mouse *M. musculus*, a key model organism for biomedical research, we identified 20 invading ERVi elements (Mmus.ERVi) that cluster into eight vOTUs (Fig. 3). In the wild, these Mmus.ERVi elements are present in 0% to 42.86% (for Mmus.ERVi.5a) of area grid cells (~12,321 km²) across the globe (Fig. 3a, e). Each ERVi is present in different prevalence among subpopulations in the wild (Fig. 3c–e, g, h). Moreover, distinct laboratory lines harbor different ERVi spectrum (Fig. 3b, f). Interestingly, five ERVi elements appear to be exclusively present in laboratory lines[14] (Fig. 3e, f). These ERVi might have arisen during the generation of laboratory mouse lines or might have been derived from unsampled wild parents, which can be resolved by further genome sequencing of wild parents.

Cichlid fishes (*Amphilophus* spp.), an extremely young species complex, are restricted to several small, isolated crater lakes (CLs) of recent origin and the great lakes (GLs) Nicaragua and Managua[15,16] (Fig. 4a). Among them, Midas cichlids in the CLs Apoyo and Xiloá are text-book examples of sympatric speciation, the formation of species without geographic barriers[15,16]. We identified three ERVi elements (Amp.ERVi) that are likely to belong to a single vOTU in the Midas cichlid fish complex. Two ERVi insertions are present in all the species with intermediate to high frequency (0.19 to 1.00 for Amp.ERVi.1a; 0.61 to 1.00 for Amp.ERVi.1b) (Fig. 4b). The remaining ERVi element (Amp.ERVi.1c) only occurred at low frequency in *A.* cf. *citrinellus* in CL Masaya and *A. citrinellus* in GL Managua and GL Nicaragua, indicative

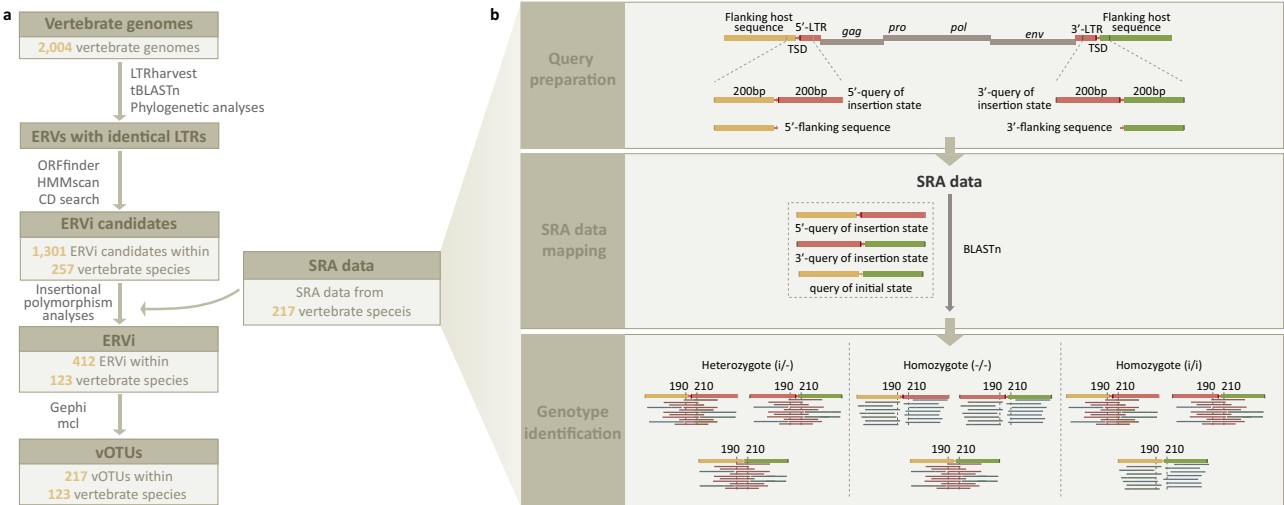

**Fig. 1 | The flowchart for ERVi identification. a** A step-by-step flowchart for ERVi identification. **b** The detailed process for insertional polymorphism analyses. ERVi LTRs were labeled in red. The 5′- and 3′-flanking regions were labeled in yellow and green, respectively. The meaningful mapped reads were labeled in dark red. See Supplementary Data 1–7 for detailed information. Abbreviations: long terminal repeat (LTR), target site duplication (TSD).

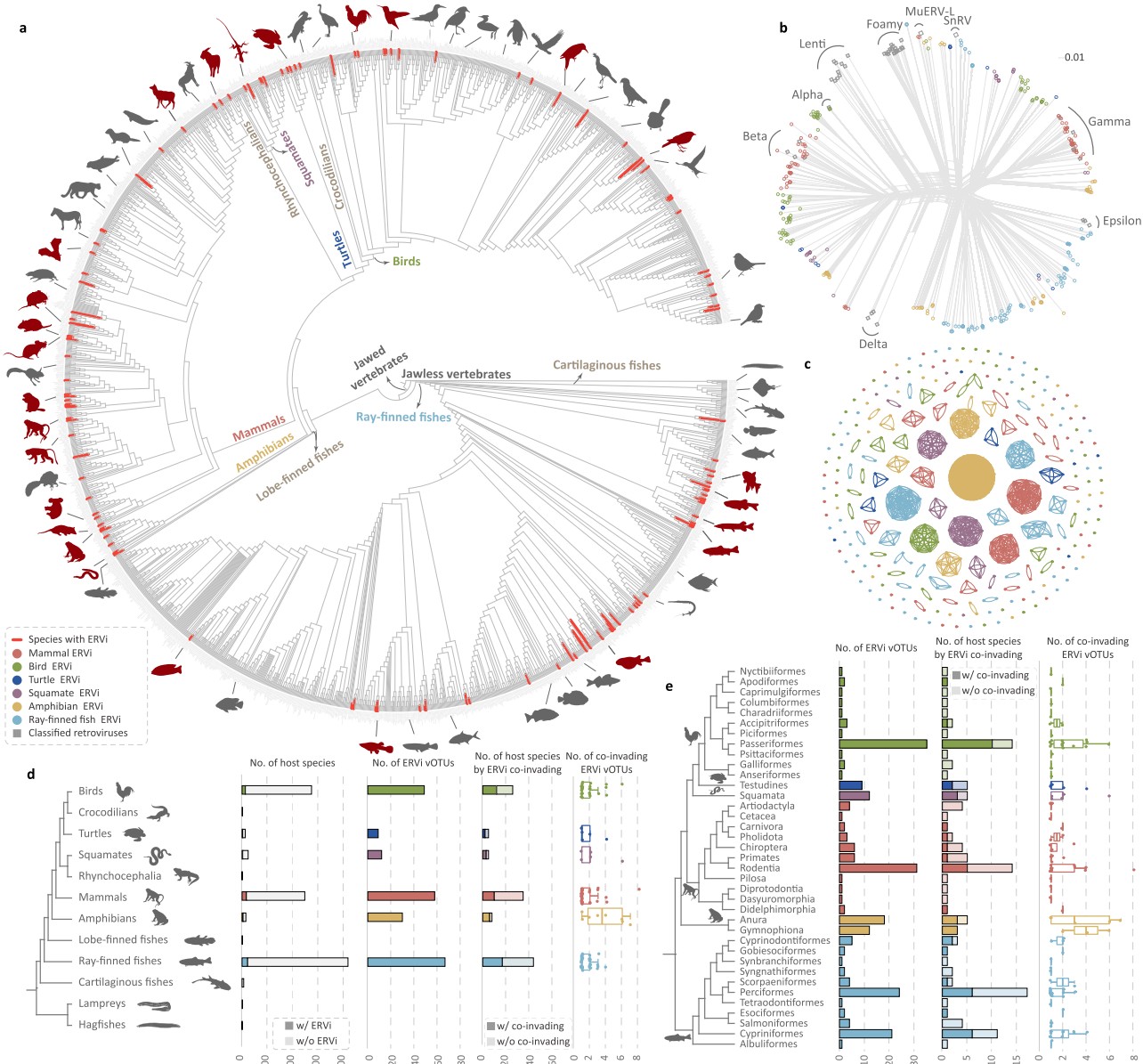

**Fig. 2 | The diversity and distribution of ERVi across vertebrates. a** Distribution of 123 vertebrates hosting ERVi across more than 2000 vertebrate species. The branches leading to species hosting ERVi are highlighted in red. The phylogeny of vertebrate species is based on TimeTree5[50] and literatures[51–58]. Four species of hybrid origin are not included in the phylogeny. **b** Phylogenetic network reconstructed based on RT proteins of 412 ERVi elements, 48 retroviruses classified by ICTV, snakehead retrovirus (SnRV), and murine endogenous retrovirus-L (MuERV-L). Filled circles and rectangles indicate ERVi and classified retroviruses, respectively. The alignment used for phylogenetic analysis is provided in Supplementary Data 11. **c** The assignment of ERVi elements into viral operational taxonomic units (vOTUs). 412 ERVi elements were grouped into 217 clusters (vOTUs) based on sequence identity. ERVi elements from distinct vertebrate clades were labeled in different colors. The identity between paired ERVi used for clustering analysis is available in Supplementary Data 6, and the results of clustering are provided in Supplementary Data 7. **d** The distribution and co-invasion pattern of ERVi in distinct vertebrate clades. The numbers of vertebrate species invaded by ERVi, ERVi vOTUs, vertebrate species invaded by multiple ERVi vOTUs, and co-invading ERVi vOTUs are shown for major vertebrate clades. The sample size for the box plot of birds, turtles, squamates, mammals, amphibians, and ray-finned fishes with co-invading ERVi vOTUs is $n = 26$, $n = 5$, $n = 5$, $n = 34$, $n = 8$, and $n = 44$, respectively. The bounds and center line of all box plots indicate the lower and upper quartiles and the

median, respectively. The minima and maxima of each box plot are also shown. Specific values are available in Supplementary Data 8. **e** The distribution and co-invasion pattern of ERVi in distinct vertebrate orders. The numbers of ERVi vOTUs, host species invaded by multiple ERVi vOTUs, and co-invading ERVi vOTUs are shown for 37 orders of vertebrates that possess species hosting ERVi. The sample size for the box plot of Nyctibiiformes, Apodiformes, Caprimulgiformes, Columbiformes, Charadriiformes, Accipitriformes, Piciformes, Passeriformes, Psittaciformes, Galliformes, Anseriformes, Testudines, Squamata, Artiodactyla, Cetacea, Carnivora, Pholidota, Chiroptera, Primates, Rodentia, Pilosa, Diprotodontia, Dasyuromorphia, Didelphimorphia, Anura, Gymnophiona, Cyprinodontiformes, Gobiesociformes, Synbranchiformes, Syngnathiformes, Scorpaeniformes, Perciformes, Tetraodontiformes, Esociformes, Salmoniformes, Cypriniformes, and Albuliformes with co-invading ERVi vOTUs is $n = 1$, $n = 1$, $n = 1$, $n = 1$, $n = 1$, $n = 2$, $n = 1$, $n = 14$, $n = 1$, $n = 2$, $n = 1$, $n = 5$, $n = 4$, $n = 1$, $n = 1$, $n = 2$, $n = 4$, $n = 5$, $n = 13$, $n = 1$, $n = 1$, $n = 1$, $n = 5$, $n = 3$, $n = 3$, $n = 1$, $n = 1$, $n = 2$, $n = 2$, $n = 17$, $n = 1$, $n = 4$, $n = 11$, and $n = 1$, respectively. The bounds and center line of all box plots indicate the lower and upper quartiles and the median, respectively. The minima and maxima of each box plot are also shown. Specific values are available in Supplementary Data 9. Silhouette images were from PHYLOPIC (https://www.phylopic.org/). Abbreviations: viral operational taxonomic units (vOTUs), with (w/) and without (w/o).

**Table 1 | D statistics for the distribution of ERVi in vertebrates**

| Repeats | Dataset | No. of host species | D | p-value (D > 0) | p-value (D < 1) |
|---|---|---|---|---|---|
| Repeat 1 | Total | 997 | 0.8169117 | 0 | 0 |
| | Mammals | 378 | 0.895802 | 0 | >0.05 |
| | Birds | 164 | 0.8081952 | <0.01 | >0.05 |
| | Reptiles | 59 | 1.335925 | 0 | >0.05 |
| | Amphibians | 28 | 0.138071 | >0.05 | <0.05 |
| | Fishes | 368 | 0.5857624 | <0.01 | <0.01 |
| Repeat 2 | Total | 983 | 0.8514632 | 0 | <0.01 |
| | Mammals | 363 | 0.9976817 | 0 | >0.05 |
| | Birds | 161 | 0.8117434 | <0.01 | >0.05 |
| | Reptiles | 59 | 1.398738 | <0.01 | >0.05 |
| | Amphibians | 28 | 0.138071 | >0.05 | <0.05 |
| | Fishes | 372 | 0.6029075 | <0.01 | <0.01 |
| Repeat 3 | Total | 828 | 0.7531927 | 0 | 0 |
| | Mammals | 315 | 0.8302291 | 0 | >0.05 |
| | Birds | 122 | 0.5512629 | >0.05 | >0.05 |
| | Reptiles | 49 | 1.033423 | <0.05 | >0.05 |
| | Amphibians | 28 | 0.1446161 | >0.05 | <0.05 |
| | Fishes | 314 | 0.5880192 | <0.01 | <0.01 |

of its more recent provenance and further supporting the infectivity of the Amp.ERVi.1 vOTU (Fig. 4b–d).

Chickens (*Gallus gallus domesticus*), the most popular domestic animals around the world, were derived from *G. g. spadiceus* (a subspecies of red jungle fowl) in the Holocene[17]. Following their initial domestication, chickens might secondarily interbreed with red jungle fowl subspecies[17]. We identified two Ggal.ERVi.1 elements in the chicken genomes (Fig. 5a, b). Both Ggal.ERVi.1 elements are present in a very limited number of domestic lines, and only Gal.ERVi.1b was identified in the population of *G. g. spadiceus* with a very low frequency of ~0.07 (Fig. 5c). These results indicate that Ggal.ERVi.1 might have begun its invasion into the genomes of *G. g. spadiceus* through secondary interbreeding after the domestication of chickens.

### Ongoing retroviral invasion in threatened vertebrates
Among the vertebrates whose genomes are invaded by ERVi, 18 species (8 mammals, 2 birds, 1 squamate, 4 turtles, 1 amphibian, and 2 ray-finned fishes) (Fig. 6 and Supplementary Fig. 8d–f) have been assessed to be threatened with extinction (5 critically endangered, 6 vulnerable, and 7 endangered) by International Union for Conservation of Nature (IUCN) red list of threatened species[8]. Phylogenetic analyses show that ERVi vOTUs hosted by threatened species do not cluster together but are dispersed across the tree (Supplementary Fig. 12). On the other hand, the proportion of species invaded by ERVi is not significantly different between threatened and non-threatened species invaded (*G*-test, *p*-value = 0.34). Together, our study indicates that retroviruses are pervasively invading the genomes of threatened vertebrates.

### Ongoing retroviral invasion in vertebrates closely associated with humans
At least 73 vertebrate species hosting ERVi are reportedly associated with humans through diverse routes (as foods, pets or display animals, medicine, research materials, wearing apparel or accessories, handicrafts or jewelry, sport hunting or specimen collecting, or establishing ex-situ production)[8] (Fig. 7). 44 and 43 species were associated with humans as foods and pets, respectively (Fig. 7a, d). Among them, 39 vertebrate species are involved in at least two association routes with humans (Fig. 7d). We predict that the frequent contact might facilitate the exposure of ERVi to humans, potentially leading to spillover and emerging infectious diseases (EIDs)[18,19].

## Discussion
In this study, we reported that diverse retroviruses are invading the genomes of more than 6% of vertebrates, revealing the unexpected prevalence of ongoing retroviral invasion in vertebrates. However, the prevalence is much underestimated, given: (i) high-quality population genomics data are only available for a small proportion of vertebrates analyzed in this study; (ii) ERVi are likely to be present in non-reference genomes[9,20,21]. Nevertheless, our study provides resources for studying the evolutionary process of the transition between exogenous retroviruses and ERVs.

ERVi we identified in this study actually represent a continuous, rather than discrete, spectrum of invading retroviruses, covering the very beginning to the relatively late stages of retroviral invasion. For instance, Marm.ERVi.1 from *Mastacembelus armatus* is likely to be at the very beginning of its invasion, because Marm.ERVi.1 is only present in one out of 22 individuals and no ERV closely related to Marm.ERVi.1 was identified in the other 21 individuals. On the other hand, ERVi identified in this study appear to be more recent invaders than human endogenous retrovirus-K (HML-2). HML-2 elements in humans probably represent the final stage of a retroviral invasion, as almost all the HML-2 elements in humans are not intact with disruptive mutations and their LTRs are not identical, even for the well-studied HERV-K113 (99.69%; with three mismatches across 968 bp) that is polymorphic in the human population and not present in human reference genome[9,20].

The recent rise of metagenomics has dramatically expanded our understanding of virus diversity and evolution[22–25]. However, retroviruses have been routinely neglected in nearly all the metagenomics analyses, because it is technically challenging to distinguish exogenous retroviruses and expressed ERVs. To date, only about 68 retroviruses have been classified by ICTV[11], severely under-representing the actual diversity of retroviruses. ERVi are viruses in transition between exogenous and endogenous status[7], and thus our study expands the diversity of retroviruses that are recently (extinct now) or currently circulating in the wild.

The ongoing retroviral invasion brings at least five-fold risks for the threatened species (and also some, if not all, for the non-threatened species): First, ERVi can produce functional viruses that are associated with diverse diseases and cause the rapid decline of their host populations[26–28]; for instance, KoRV-A is associated with lymphoma and leukemia in koalas[7,29,30]. Second, insertion of ERVi into or near crucial host genes might lead to genetic diseases[31]. Third, recombination between ERVi can mediate chromosome rearrangements, resulting in genomic instability[32,33]. Fourth, ERVi might recombine with other circulating retroviruses or with other ERVs, generating viruses with new pathogenicity[34]. Five, weakly deleterious ERVi insertions might not be effectively removed from their host populations due to the low population size of the threatened species. Understanding the pathogenicity of ERVi is of great relevance to the conservation of threatened vertebrates.

EIDs are increasingly threatening to global public health[19]. The majority (~71.8%) of EIDs, as exemplified by the emergence of AIDS and Ebola, were caused by the spillover of zoonotic pathogens in wildlife via frequent contact at the human-animal interface[18,19]. The discovery of hundreds of ERVi that are at least recently active at the human-animal interface also provides resources for virus surveillance and zoonotic risk assessment[35].

## Methods
### The criteria to identify ERVi
We used the following criteria to attribute an ERV to an ERVi: First, its flanking LTRs are identical at the nucleotide level; Second, its encoding genes (*gag*, *pro*, *pol*, and *env*) are intact without disruptive mutations; Third, it is insertionally polymorphic among individuals in the host population (Fig. 1).

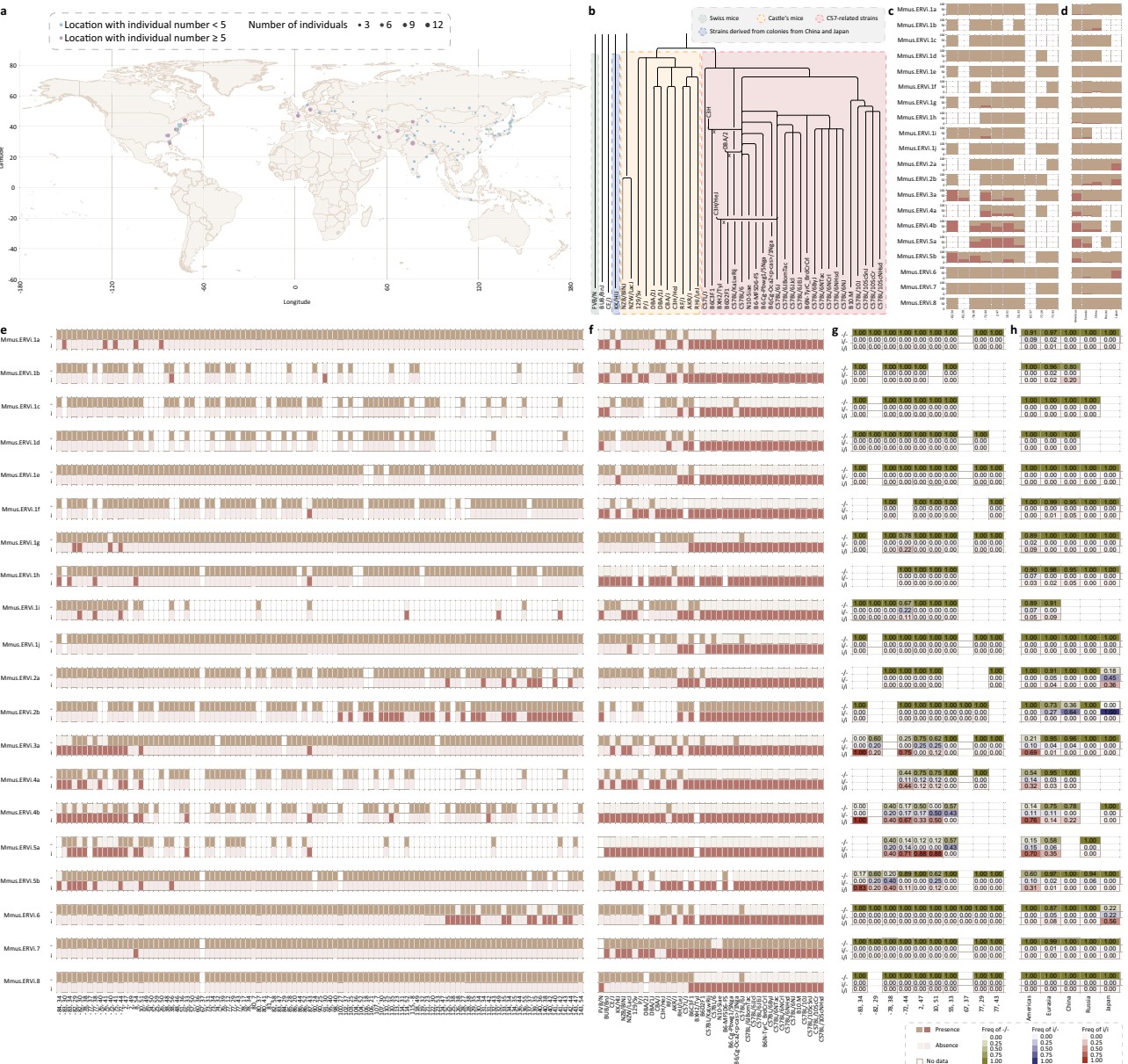

**Fig. 3 | The geographic distribution and prevalence of Mmus.ERVi. a** The geographic distribution of sampling locations for mice in their natural habitats. Circle size indicates sample size in the corresponding location. Locations with individual numbers < and ≥5 are labeled in blue and purple, respectively. **b** Genealogies of inbred mice used in this study[14]. Inbred strains of different backgrounds are labeled in different colors. **c** The frequency of Mmus.ERVi insertion (i) and empty locus (−) for wild mice in different area grid cells (-12,321 km²). **d** The frequency of Mmus.ERVi insertion (i) and empty locus (−) for wild mice in different geographic regions. **e** The presence and absence of Mmus.ERVi for wild mice in different area grid cells (-12,321 km²). **f** The presence and absence of Mmus.ERVi in different laboratory strains. **g** The genotype (i/i, i/−, and −/−) frequency of Mmus.ERVi for wild mice in different area grid cells (-12,321 km²). **h** The genotype (i/i, i/−, and −/−) frequency of Mmus.ERVi for wild mice in different geographic regions. For an area grid cell (-12,321 km²) or a geographic region, when population genomics data for more than 4 individuals are available, the frequency of genotypes (i/i, i/−, and −/−) as well as the frequency of ERVi insertion (i) and empty locus (−) were shown.

## Identification of ERVs with identical LTRs

All the representative genomes of 2004 vertebrates were retrieved from NCBI (https://www.ncbi.nlm.nih.gov/genome/). We used a similarity search and phylogenetic analysis combined approach to identify ERVs in the genomes of 2004 vertebrates (-2.783 tetra-bases), including 494 mammals, 547 birds, 78 reptiles (4 crocodilians, 25 turtles, 48 squamates, and 1 rhynchocephalians), 30 amphibians, 3 lobe-finned fishes, 834 ray-finned fishes, 13 cartilaginous fishes, 4 lampreys, and 1 hagfish (Fig. 1 and Supplementary Data 1). LTRharvest implemented in GenomeTools v1.5.10 was used to identify genetic elements with identical 5'- and 3'-LTRs by setting -similar to 100 [36,37]. Then homologs of retrovirus reverse transcriptase (RT) proteins were identified from the elements with identical 5'- and 3'-LTRs using the tBLASTn algorithm implemented in BLAST 2.12.0+ with RT protein sequences of 10 representative retroviruses as queries and an *e* cut-off value of 10⁻⁵ (Supplementary Data 2). Significant hits longer than 100 amino acids (aa) and without any premature stop codons were retrieved for phylogenetic analyses. Because retrotransposons and retroviruses share certain sequence similarities in their RT proteins, phylogenetic analyses of the RT protein sequences from significant hits, representative retrotransposons, and representative retroviruses were performed to identify sequences that cluster with retrovirus RT proteins, that is, ERVs[38]. Sequences were aligned using MAFFT v7.402 with the L-INS-I strategy[39]. An approximate maximum-likelihood (ML)

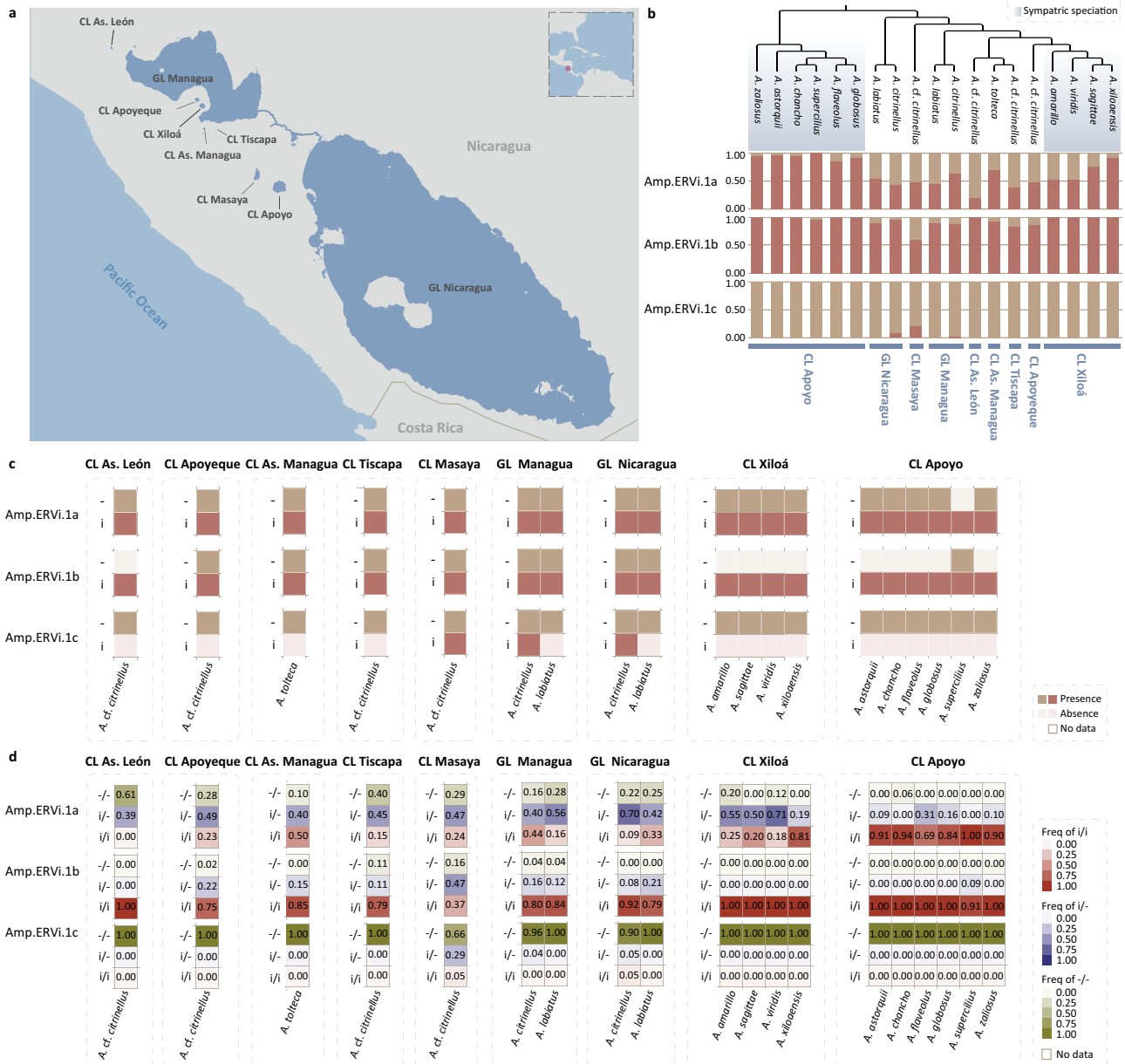

**Fig. 4 | The geographic distribution and prevalence of Amp.ERVi. a** The geographic distribution of sampling locations for Midas cichlids. The map was created using MAPCREATOR (https://api.mapcreator.io/). **b** The frequency of Amp.ERVi insertion (i) and empty locus (−) in different populations. Phylogeny of Midas cichlids is based on literature[16]. **c** The presence and absence of Amp.ERVi in different populations. **d** The genotype (i/i, i/−, and −/−) frequency of Amp.ERVi in different populations. Only populations with individual numbers of >4 are used in (**d**). Abbreviations: crater lake (CL), great lake (GL).

method implemented in FastTree v2.1.10 was used to perform the initial phylogenetic analyses[40].

### Identification of ERVi candidates

We analyzed the characteristics of genome structures and open reading frames (ORFs) for all the 48 replication-competent retroviruses classified by ICTV[11] (Supplementary Data 3). Then, we used ORFfinder v0.4.3 to predict ORFs for each ERV with standard genetic codes and annotated putative *gag*, *pro*, *pol*, and *env* genes (Fig. 1). 80% of the shortest length for each core ORF among the 48 classified retroviruses was used as the cut-off for intact ORFs (Fig. 1 and Supplementary Data 3). The ERV genomes with *gag-pro-pol-env* genome structures, with RT-RH-IN domain architecture in Pol protein, tm motif in Env protein, and without any premature stop codons in putative ORFs were retrieved (Fig. 1). Conserved domain (CD) search[41] against Conserved Domain Database version 3.20 and HMMscan[42]

implemented in HMMER 3.2.1 were used to annotate the domain architectures for ERVs with the default parameters. We identified a total of 1301 ERVi candidates within 257 vertebrate species (Fig. 1).

### Insertional polymorphism analyses

To further confirm that these ERVi candidates are invading their host genomes, we performed insertional polymorphism analyses using almost all the available genome-scale sequence read archive (SRA) data (4,316) from 217 host species in NCBI (~168.12 tera-bases in total) (Fig. 1 and Supplementary Data 4). For species (e.g. *M. musculus*) with too many public SRA data available in NCBI, we randomly selected some SRA data to perform the insertional polymorphism analyses for them. First, each ERVi candidate was bidirectionally extended to 200 nucleotides. Then, these 5′- and 3′-flanking regions for each ERVi candidate were joined together as a "query of initial state" by removing one target site duplication (TSD) (Fig. 1). "Query of initial state"

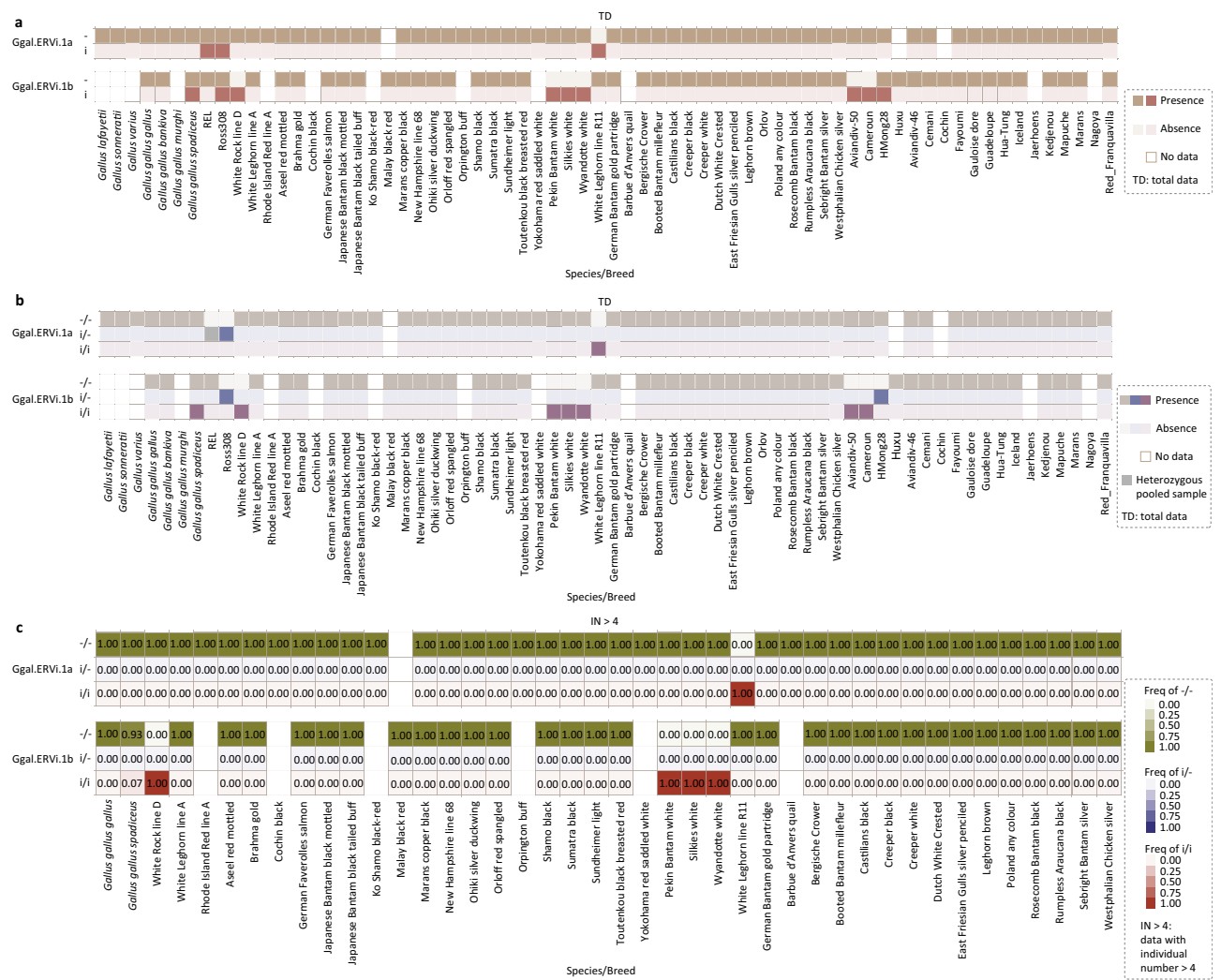

**Fig. 5 | The prevalence of Ggal.ERVi. a** The presence and absence of Ggal.ERVi in jungle fowls and different breeds of *Gallus gallus*. **b** The presence and absence of different genotypes (i/i, i/−, and −/−) for Ggal.ERVi in jungle fowls and different breeds of *G. gallus*. Genome sequencing data of pooled individuals without specified sample sizes that can be inferred to be heterozygous (i/−) are labeled in gray. **c** The frequency of Ggal.ERVi insertion (i) and empty locus (−) in jungle fowls and different breeds of *G. gallus*. TD (total data) include all the evidentiary genome sequencing data of individuals, pooled individuals with or without sample size, and samples without detailed individual information. For each breed, when population genomics data for more than 4 individuals (IN > 4) are available, the frequency of genotypes (i/i, i/−, and −/−) as well as the frequency of ERVi insertion (i) and empty locus (−) were shown. The IN > 4 dataset is composed of genome sequencing data of individuals and pooled individuals with a sample size that can be inferred to be homozygous (i/i or −/−).

represents the ancestral state without ERVi insertion (Fig. 1). Simultaneously, 5′- and 3′- "queries of insertion state" reflecting the ERV insertion state were also generated (Fig. 1). The BLASTn algorithm implemented in BLAST 2.12.0+ was used to map raw reads on to the three queries (query of the initial state as well as 5′- and 3′-queries of insertion state) with an *e* cut-off value of $10^{-5}$ and an identity cut-off value of 99% (Fig. 1). A read is considered to be meaningful, only when it is full-length mapped and its map region fully covers the positions 190–210 of queries (the junction region) (Fig. 1). Considering the difference of sequencing coverage, one meaningful read was used as a cut-off to identify the presence of ERVi candidates in the host genomes. SRA data that provide evidence for insertional polymorphism of ERVi were considered evidentiary sequencing data. Only ERVi candidates with evidence of insertional polymorphism are considered to be authentic ERVi (Fig. 1). We identified a total of 412 ERVi elements in the genomes of 123 vertebrate species (Fig. 1 and Supplementary Data 5).

To more accurately and fully identify insertional polymorphism (i, locus with ERVi insertion; -, empty locus without ERVi insertion), diverse genome-scale datasets were used. TD (total data) includes all the evidentiary genome sequencing data of individuals, pooled individuals with or without specified sample size, and samples without detailed individual information. ID (individual data) includes evidentiary genome sequencing data for individuals and pooled individuals with specified sample size that can be inferred to be homozygous (i/i or −/−); LD represents evidentiary genome sequencing data with location information. For an area grid cell (-12,321 km²) or a geographic region, when population genomics data for more than 4 individuals (IN > 4) are available, the frequency of genotypes (i/i, i/−, and −/−) as well as the frequency of ERVi insertion (i) and empty locus (−) were inferred. Moreover, evidentiary sequencing data of pooled individuals or samples without detailed individual information that can be inferred to be homozygous (i/i or −/−) were also used to reveal the presence and absence of genotypes (i/i, i/−, and −/−). Evidentiary sequencing data of pooled individuals with a specified sample size that can be inferred to be homozygous (i/i or −/−) were also used to calculate the frequency of genotypes (i/i, i/−, and −/−) as well as the frequency of ERVi insertion (i) and empty locus (−).

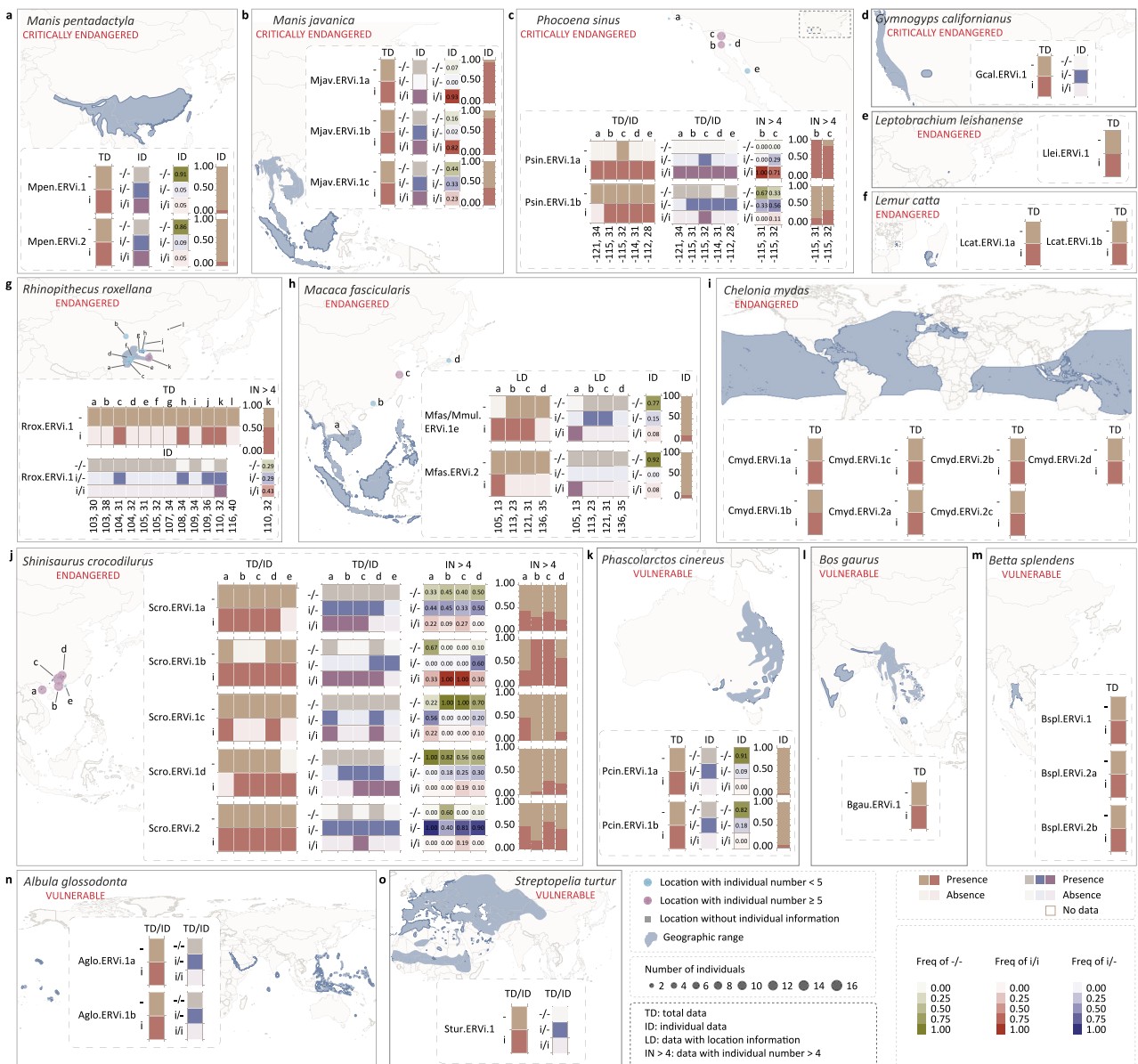

**Fig. 6 | The geographic distribution and prevalence of ERVi in 15 threatened species [International Union for Conservation of Nature (IUCN) red list category: critically endangered (CR), Endangered (EN), and Vulnerable (VU)].**
**a** *Manis pentadactyla* (category: CR). **b** *Manis javanica* (category: CR). **c** *Phocoena sinus* (category: CR). **d** *Gymnogyps californianus* (category: CR). **e** *Leptobrachium leishanense* (category: EN). **f** *Lemur catta* (category: EN). **g** *Rhinopithecus roxellana* (category: EN). **h** *Macaca fascicularis* (category: EN). **i** *Chelonia mydas* (category: EN). **j** *Shinisaurus crocodilurus* (category: EN). **k** *Phascolarctos cinereus* (category: VU). **l** *Bos gaurus* (category: VU). **m** *Betta splendens* (category: VU). **n** *Albula glossodonta* (category: VU). **o** *Streptopelia turtur* (category: VU). The geographic range for each species was retrieved from IUCN red list of threatened speices[8] and shown in blue. For each species, diverse datasets were used to identify the presence (i) and absence (−) of ERVi insertion in a locus: TD (total data) includes all the evidentiary genome sequencing data of individuals, pooled individuals with or without specified sample size, and samples without detailed individual information; ID (individual data) include evidentiary genome sequencing data for individuals or pooled individuals with specified sample size that can be inferred to be homozygous (i/i or −/−); LD represents evidentiary genome sequencing data with location information. LD data used in (**h**) include multiple samples with location information but without individual information, from which homozygous genotypes (i/i or −/−) can be inferred. For each species with ID or LD data, genotypes i/i, i/−, and −/− were shown. For an area grid cell (−12,321 km²), when population genomics data for more than 4 individuals (IN > 4) are available, the frequency of genotypes (i/i, i/−, and −/−) as well as the frequency of ERVi insertion (i) and empty locus (−) were shown. Circle size indicates sample size in the corresponding location. Locations with individual numbers < and ≥5 are labeled in blue and purple, respectively. Locations without detailed individual information are labeled with gray rectangles.

## Classification of ERVi

We classified ERVi into vOTUs based on their genome sequences using the criteria: sequence coverage of ≥80%, sequence identity of ≥95%, and e-value of ≤1 × 10⁻⁵[22,23,43]. These criteria are strict enough to divide all the 48 replication-competent retroviruses classified by ICTV into individual retrovirus vOTU[11]. Two different algorithms [Markov CLuster (MCL) algorithm[44] and community detection algorithm[45]] were used to cluster ERVi into individual vOTU. The nucleotide identity of paired ERVi was estimated using all-to-all BLASTn algorithm (-qcov_hsp_perc 80 -perc_identity 95 -evalue 0.00001). 412 ERVi elements were classified into 217 ERVi vOTUs (Fig. 1, Fig. 2c and Supplementary Data 5). Gephi v0.9.7[46] with default parameters and mcl 14.137[44] with option -I 1.4 was used to clustering ERVi.

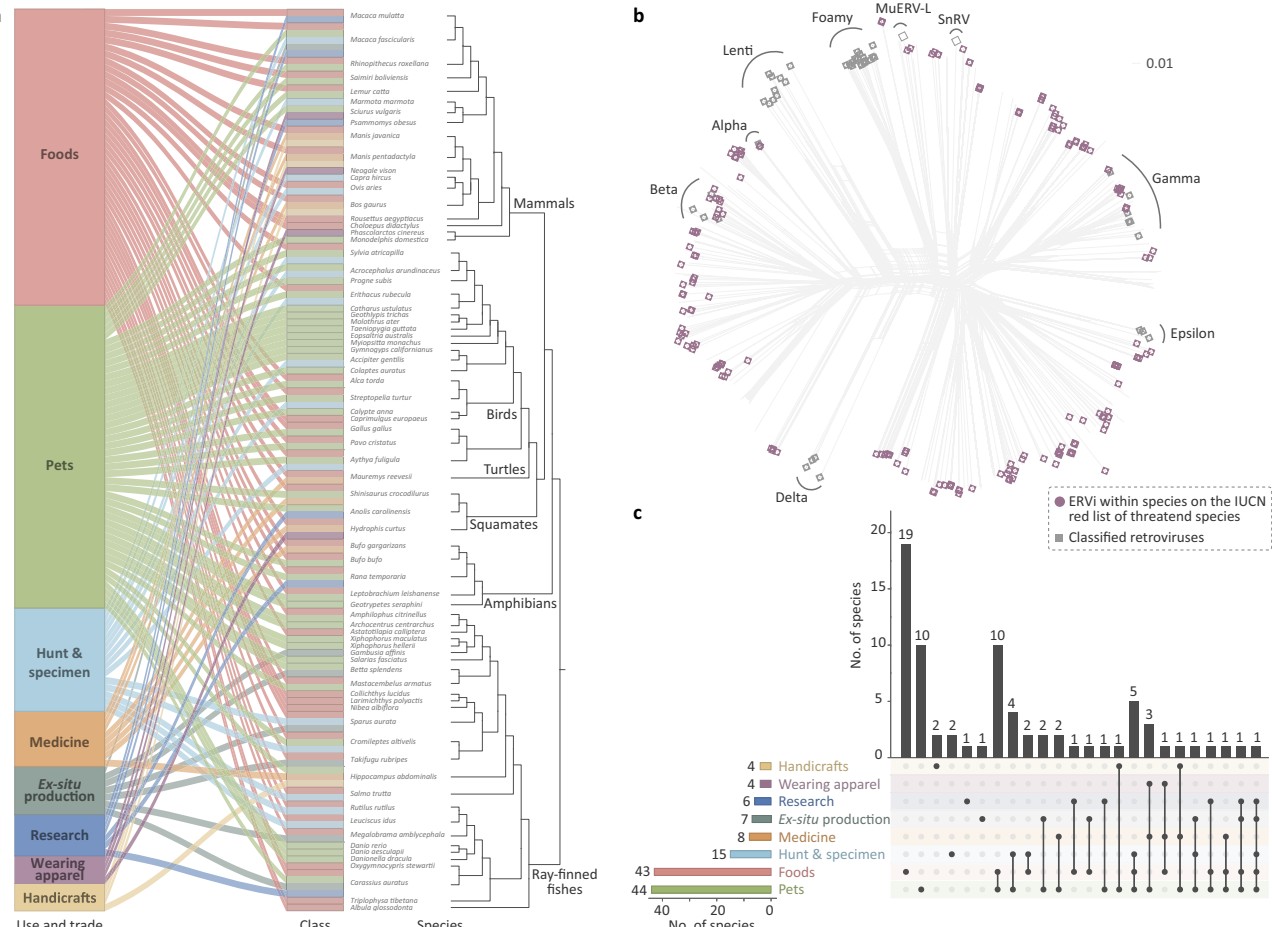

**Fig. 7 | ERVi at the interface between vertebrates and humans. a** The reported association routes for the contact between vertebrates and humans are based on the International Union for Conservation of Nature (IUCN) red list of threatened species[8]. The human-vertebrate association and species hosting ERVi are linked. **b** The diversity of ERVi within vertebrate species closely related to human daily life. These ERVi are labeled with purple filled circles. **c** The upset plot of vertebrate species with ERVi closely related to human daily life. Data for (**a**) and (**c**) are available in Supplementary Data 10. Abbreviations: snakehead retrovirus (SnRV); murine endogenous retrovirus-L (MuERV-L); pets or display animals (Pets); sport hunting or specimen collecting (Hunt & specimen); establishing ex-situ production (Ex-situ production); wearing apparel or accessories (Wearing apparel); handicrafts or jewelry (Handicrafts).

To identify orthologs or paralogs of ERVi insertions, we bidirectionally extended 500 bp for each ERVi. The 5′- and 3′-LTR with their 500 bp flanking regions were used as queries to identify orthologous or paralogous ERVi within the genomes of the corresponding species with the BlastN algorithm implemented in BLAST 2.12.0+ (-qcov_hsp_perc 100 -perc_identity 95 -evalue 0.00001).

**Phylogenetic network reconstruction of ERVi**
The RT protein sequences of 412 ERVi, 48 retroviruses classified by ICTV[11], snakehead retrovirus (SnRV), and murine endogenous retrovirus-L (MuERV-L) were aligned using the L-INS-I strategy in MAFFT v7.475[39] (Supplementary Data 3, Data 5 and Data 11). Alignment was refined using trimAl v1.2 with option -automated1[47]. SplitsTree v4.18.3 was used to reconstruct the phylogenetic network with default parameters[48].

**Simulation**
A bootstrapping simulation was performed using R v4.2.1 to estimate the total number of vOTUs in 74,140 currently described vertebrates[8]. The seed value was set to 1234. The sample_n function implemented in dplyr package v1.1.2 was used to generate random samples to estimate total vOTUs for different sample sizes. The boot package v1.3.28.1 was used to generate bootstrap samples based on 10,000 replicates. The boot.ci[49] function with the adjusted bootstrap percentile (BCa)

method implemented in boot package v1.3.28.1 was used to calculate 95% confidence intervals for each sample size.

***D* statistics**
*D* statistics is a robust measure of the phylogenetic signal of a specific binary trait. When the sample size of species is >50, *D* is independent of the prevalence of traits and the size and shape of phylogenies[13]. The phylo.d[13] function implemented in caper package v1.0.2 was used to estimate *D* statistics. Because of the zero length for some internal branches, we randomly selected host species to generate host phylogenies with non-zero branch lengths and performed three replicates. The phylogenetic tree was retrieved from TimeTree5[50].

***G*-test**
*G*-test of independence was performed using G.test function under RVAideMemoire package v0.9.83 in R.

**Reporting summary**
Further information on research design is available in the Nature Portfolio Reporting Summary linked to this article.

## Data availability
No new data were generated in support of this research. All data used or generated in this study were provided in Supplementary Data or

repository. Vertebrate genomes and SRA data used in this study were retrieved from NCBI, and the accession numbers were provided in Supplementary Data 1 and 4, respectively. The sequences of all the ERVi identified in this study have been deposited in Figshare (https://doi.org/10.6084/m9.figshare.23653254). Sequence alignment of RT protein sequences for phylogenetic analysis is provided as Supplementary Data 11.

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

## Acknowledgements

This work was supported by National Natural Science Foundation of China (32270684 and 31922001). Genome sequencing data of *Capra hircus* are used with permission from the VarGoats project. Unpublished sequencing data for *Phascolarctos cinereus*, *Eopsaltria australis*, *Manis javanica*, *Manis pentadactyla*, *Cavia porcellus*, and *Sciurus carolinensis* are used with permission from the DNA Zoo Consortium (dnazoo.org).

## Author contributions

G.-Z.H. designed research; J.W. performed research; J.W. and G.-Z.H. analyzed data; G.-Z.H. and J.W. wrote the paper.

## Competing interests

The authors declare no competing interests.
