## [Peer Review File · Nature Communications]

REVIEWER COMMENTS

Reviewer #1 (Remarks to the Author):

This manuscript reports the discovery of many insertionally polymorphic ERVs, across hundreds of different vertebrate host species. Insertionally polymorphic ERVs are present at a particular location/allele in one host genome but are absent from the host genome of another member of the host population or species.

Line 41/42 perhaps implies that the examples presented here are like the previously reported Koala retrovirus. That is to say that a new lineage of ERVs have started invading the host genomes/species reported- some individuals do not contain any members of the ERV lineage whereas others contain one or many elements. Thus, the ERV invasion in Koalas must be very recent (as must the original cross species transmission event of the initial retrovirus).

As it makes clearer later in the report this is probably not the case here. This paper likely reports instead the presence of insertionally polymorphic members of an already established ERV lineage. These are already known to be quite common. For example, all human individuals contain numerous insertionally polymorphic HERV.K (HML2) elements along with many hundreds of fixed elements belonging to the same lineage. Thus HERV.K (HML2) is certainly not the result of a recent cross-species transmission event as the family has likely been invading primate genomes for at least 30 million years. Many other examples are known of this type of insertionally polymorphic (but likely ancient) lineage. Furthermore, excluding highly inbred populations, such ancient lineages are unlikely to be highly pathogenic in the host species.

The authors need to be clearer on the type of invasion they are describing here with those akin to the Koala retrovirus being more interesting (in my opinion) than the HERV.K (HML2) example. The paper however does drastically increase the number of known insertionally polymorphic ERV lineages in various species, which is a very interesting, although not entirely unexpected, result.

Reviewer #2 (Remarks to the Author):

Re: Review of manuscript "Retroviruses are pervasively invading vertebrate genomes" as submitted by Jianhua Wang and Guan-Zhu Han.

Nature Communications NCOMMS-23-01891

Summary:

In this manuscript, the authors perform a comprehensive survey 2,004 genomes for retrovirus discovery/presence using LTRharvest under criteria of: 1. Identical flanking LTRs; 2. ORFs corresponding to gag, pro, pol, and env were present; 3. Insertional polymorphism among its host population as inferred by genotyping of recapitulated empty and 5' and 3' LTR-flank junctions against whole-genome reads from the SRA. Steps one and two resulted in identification of 1,301 ERV loci candidates (over 257 species) that were curated to a set of 412 insertions (123 species) in step three. The set of 412 ERV could be further classified into 217 viral operational taxonomic units (vOTUs). Across major lineages of vertebrates (mammals, birds, reptiles, amphibians, fishes), ERV appeared to be enriched by D statistic, suggesting elevated germline invasion in amphibians and fish versus reptiles, birds, and mammals. The findings highlight retrovirus invasion in the genomes of threatened (e.g., endangered, critically endangered) species as well as those closely associated to humans (e.g., pets, food, other). Biogeographical analysis highlights recent ERV-host dynamics.

The results are noteworthy in adding to our understanding of recently circulating retroviruses in global species. The reviewer sees the study as a valuable resource for comprehensive studies addressing virus-host evolutionary patterns and dynamics as well as possible emergence of recombinant virus types. The methodology appears sound.

Comments and Concerns:

Major Concerns:

The authors reference the analysis of “terabase-scale” or “TB scale” corresponding to 2,004 genomes over several areas of the manuscript text (e.g., lines 12; 44; 54, “>100 TB”; 183, “~2.783 TB”). What data corresponds to “>100TB”? Is this including read data from the SRA? Reference to terabyte data? Suggest moving the accurate value TB reference to Materials and Methods only.

Lines 61-62: please clarify “expand from amplification accompanying invasion”. Is this stated to infer multiple instances of germline invasion by related ERVs? In this regard, a missing parameter is exclusion of ERV copies that may be present by recent chromosomal duplication. Have this been addressed in curating the 412 ERVi loci?

Lines 34-36: Missing information, particularly in the context of ERV replication modes. Complementation in trans includes co-packaging of defective or mildly defective RNAs that results in their complementation following reverse transcription. This scenario can lead to infectious virions and may lead to breakout. Please edit the text to address. In this regard (but perhaps not in the scope of the present study), the authors have data in-hand to address ERV copies resulting from such a scenario, for example, phylogenetic analysis of viral genes and comparison of branching patterns and gene groupings. As such the study lends a robust resource for future examination of contributions of recombinants to the ERVi pool as well as other analyses concerning co-evolution between host and virus.

What is meant by Repeat 1, 2, and 3 in Table 1?

Line 65: ERVi are endogenous; the source lineage may be in transition to colonizing the germline, for example from multiple infection events, the detection of which many may be inherently non-discoverable due to limitations in genome data. Suggest removal of “essentially” and clarification. This point is highlighted elsewhere throughout the text. For example, line 163: “might recombine with other circulating retroviruses, generating viruses exhibiting new pathogenicity”. Recombinants with altered pathogenicities may also be formed via co-packaged ERV copies.

Line 155: “...and elongates the list of retroviruses being circulating in the wild...”. The study does indeed add to our knowledge of retrovirus diversity, but the reviewer suggest caution on the authors’ statement that the data represent ERVs with counterparts that are “being circulated” as the data do not confirm current infectious viruses but assesses a captured endogenous copy. This point raises concern of the discrepancy of the difficulties in age estimates of the ERVi discovered and characterized by the authors. Though it is prudent to infer the most recent integrants as having identical LTRs (as well as ORFs and unfixed status), estimates of formation could be loosely inferred by LTR-LTR comparison, and because this approach is limited to the host neutral mutation rate the observations of 0 changes between LTRs is still limited to estimates at least in the hundreds of thousands of years. Having comparison of orthologous loci may narrow this range, as well as genotyping to determine a ‘degree’ of insertion polymorphism, but these findings can not be used to confirm the presence of contemporary viruses still circulating among the surveyed populations. To clarify for the audience, please edit this statement to reflect inference of recently and/or putatively ERVi relatives in those populations.

Minor Concerns:

Line13 suggest: “... and show that at least 412 ERVi representing 217 viral operational taxonomic units...”

The Introduction requires some streamlining and addressed concerns:

Line 26: suggest “...integrated proviruses may become vertically...”, or “have the potential to”

Line 28: suggest removing "Serially documenting the past retroviral infections" and perhaps replacing with "Replicated as a permanent part of the host genome"

Lines 38: please clarify "in the early phases"

Line 42: suggest "... rarely documents, however examples exist such as the Koala..."

Line 39: suggest removing the sentences "Meanwhile, various disruptive mutations begin to occur in ERVs"

Line 43: suggest "...we performed phylogenomic mining..."

Line 44: The authors state more than once "over" 2,004 genomes; please remove for specificity of analyses using the 2,004 versus additional analysis of SRA data for screening of insertional polymorphism.

Line 47: suggest "...vertebrates and expands our knowledge of the diversity"

Line 66: suggest removing "the", per "circulating in the vertebrates"

Line 77: suggest add reference PMID: 27001843 (Halo et al, 2019)

Line 140: please define EIDs (line 168)

Line 150: suggest add reference PMID: 25535393 (Hayward et al, 2014)

Reviewer #3 (Remarks to the Author):

This manuscript would seem to make a compelling case that (a) germ-line infections by retroviruses are much commoner than previously shown and (b) that the diversity of retroviruses is much greater than previously thought. These are important conclusions and I think both are correct. However, in some respects the manuscript tends to resemble an extended abstract. There is almost too much information included here and perhaps too little description of how it was put together or enough consideration given to the significance of qualitative differences between the different viruses. In addition, there seemed to be a tendency to state results rather than to show and interpret data leading to a conclusion. Moreover, it was sometimes hard to find the information needed to evaluate all the statements made. There are several areas which I would like to see addressed in more detail.

First, is it possible that some of the new proviruses in mice do not reflect new genomic invasions but rather the result of amplification of ERVs which are already fixed in the population and thus not considered an ERVi? The clusters of new ERVs in *M. mus* (Mmus.ERVi.1a-j in F3) and *M. spr* (Mspr.ERVi.3a-g in F3S) would seem consistent with such a view. How different are they from one another – are alignments readily available? It might be worth assembling a list of intact ERVs that do not show insertional polymorphisms – sequence comparisons with the ERVi list might shed light on the latter's origins.

Second, considering the *Mus* viruses it is worth pointing out that many would not consider *Musculus* one distinct species – are all 8 vOTUs equally distributed in *domesticus/musculus/castaneus*? Might some of the apparent invasion events during the generation of laboratory mice be explained if one or more their parents were not included in the random sample?

Third, I wonder whether the diversity of retroviruses is not altogether unanticipated. See Gifford et al *Retrovirology* 15:59, 2018 for a discussion of some of the difficulties in establishing a unified ERV/retrovirus phylogeny. It would be interesting to see a comparison between the tree in F2 of the current manuscript with that of F4 in the proceeding paper with its suggestion of "three 'placeholder' groups designed to act as temporary 'bins' for ERV loci that cannot be confidently placed within the existing taxonomic system approved by the ICTV". Again, it would be nice to

have easily accessible protein alignments.

RESPONSE TO REVIEWERS' COMMENTS

Reviewer #1 (Remarks to the Author):

This manuscript reports the discovery of many insertionally polymorphic ERVs, across hundreds of different vertebrate host species. Insertionally polymorphic ERVs are present at a particular location/allele in one host genome but are absent from the host genome of another member of the host population or species.

Line 41/42 perhaps implies that the examples presented here are like the previously reported Koala retrovirus. That is to say that a new lineage of ERVs have started invading the host genomes/species reported- some individuals do not contain any members of the ERV lineage whereas others contain one or many elements. Thus, the ERV invasion in Koalas must be very recent (as must the original cross species transmission event of the initial retrovirus).

As it makes clearer later in the report this is probably not the case here. This paper likely reports instead the presence of insertionally polymorphic members of an already established ERV lineage. These are already known to be quite common. For example, all human individuals contain numerous insertionally polymorphic HERV.K (HML2) elements along with many hundreds of fixed elements belonging to the same lineage. Thus HERV.K (HML2) is certainly not the result of a recent cross-species transmission event as the family has likely been invading primate genomes for at least 30 million years. Many other examples are known of this type of insertionally polymorphic (but likely ancient) lineage. Furthermore, excluding highly inbred populations, such ancient lineages are unlikely to be highly pathogenic in the host species.

The authors need to be clearer on the type of invasion they are describing here with those akin to the Koala retrovirus being more interesting (in my opinion) than the HERV.K (HML2) example. The paper however does drastically increase the number of known insertionally polymorphic ERV lineages in various species, which is a very interesting, although not entirely unexpected, result.

Response: Thanks for the appreciation and all these great comments. ERV_i we identified in this study actually represent a continuous, rather than discrete, spectrum of invading retroviruses, covering the very beginning to the relatively late stages of retroviral invasions. Koala retrovirus A might reflect the early stage of a retroviral invasion. On the other hand, HML-2 elements in human are likely to represent the final stage of a retroviral invasion, as almost all the HML-2 elements in human are not intact with disruptive mutations and their LTRs are not identical (even for the well-studied HERV-K113 (99.69%; with three mismatches), which is not present in the human reference genome). Therefore, ERV_i identified in this study is likely to be more active and recent than HML-2 elements. We clarified this in the Discussion section of our revised manuscript.

Reviewer #2 (Remarks to the Author):

Re: Review of manuscript “Retroviruses are pervasively invading vertebrate genomes” as submitted by Jianhua Wang and Guan-Zhu Han.

Nature Communications NCOMMS-23-01891

Summary:

In this manuscript, the authors perform a comprehensive survey 2,004 genomes for retrovirus discovery/presence using LTRharvest under criteria of: 1. Identical flanking LTRs; 2. ORFs corresponding to gag, pro, pol, and env were present; 3. Insertional polymorphism among its host population as inferred by genotyping of recapitulated empty and 5' and 3' LTR-flank junctions against whole-genome reads from the SRA. Steps one and two resulted in identification of 1,301 ERV loci candidates (over 257 species) that were curated to a set of 412 insertions (123 species) in step three. The set of 412 ERV could be further classified into 217 viral operational taxonomic units (vOTUs). Across major lineages of vertebrates (mammals, birds, reptiles, amphibians, fishes), ERV appeared to be enriched by D statistic, suggesting elevated germline invasion in amphibians and fish versus reptiles, birds, and mammals. The findings highlight retrovirus invasion in the genomes of threatened (e.g., endangered, critically endangered) species as well as those closely associated to humans (e.g., pets, food, other). Biogeographical analysis highlights recent ERV-host dynamics.

The results are noteworthy in adding to our understanding of recently circulating retroviruses in global species. The reviewer sees the study as a valuable resource for comprehensive studies addressing virus-host evolutionary patterns and dynamics as well as possible emergence of recombinant virus types. The methodology appears sound.

Response: Thanks for the great and detailed summary!

Comments and Concerns:

Major Concerns:

The authors reference the analysis of “terabase-scale” or “TB scale” corresponding to 2,004 genomes over several areas of the manuscript text (e.g., lines 12; 44; 54, “>100 TB”; 183, “~2.783 TB”). What data corresponds to “>100TB”? Is this including read data from the SRA? Reference to terabyte data? Suggest moving the accurate value TB reference to Materials and Methods only.

Response: Done as suggested! In the revised manuscript, we moved the accurate value to Material and Methods.

Lines 61-62: please clarify “expand from amplification accompanying invasion”. Is this stated to infer multiple instances of germline invasion by related ERVs? In this regard, a

missing parameter is exclusion of ERV copies that may be present by recent chromosomal duplication. Have this been addressed in curating the 412 ERVi loci?

Response: Thanks for this great comment. The statement “expand from amplification accompany invasion” means that ERV amplifies itself within its host genome through reinfection or retrotransposition either in *cis* or in *trans*. To explore the possibility of recent chromosomal duplication raised by this reviewer, we bidirectionally extended 500bp for each ERVi and identified ERV copies generated through recent chromosomal duplication within the genome of corresponding species. We found that only Mate.ERV.2a and Mate.ERV.2b were derived from recent chromosomal duplication. Therefore, it seems reasonable to state that “elements within a single vOTU are largely derived from amplification accompanying invasion”. We also added the case of recent duplication (Mate.ERV.2a and Mate.ERV.2b) in the revised manuscript.

Lines 34-36: Missing information, particularly in the context of ERV replication modes. Complementation in trans includes co-packaging of defective or mildly defective RNAs that results in their complementation following reverse transcription. This scenario can lead to infectious virions and may lead to breakout. Please edit the text to address. In this regard (but perhaps not in the scope of the present study), the authors have data in-hand to address ERV copies resulting from such a scenario, for example, phylogenetic analysis of viral genes and comparison of branching patterns and gene groupings. As such the study lends a robust resource for future examination of contributions of recombinants to the ERVi pool as well as other analyses concerning co-evolution between host and virus.

Response: Thanks for raising this complementation scenario. We added the scenario of co-packaging of defective RNAs in the Introduction section. Overall, recombination is very complex, and it is challenging to decipher the actual parents and history for a recombination event. In this regard, it is infeasible to tell whether recombination occurred in the progenitor of ERVi in the deep past or recombination among defective ERVs generated ERVi. We agree this might not be the scope of the present study.

What is meant by Repeat 1, 2, and 3 in Table 1?

Response: As stated in the manuscript, “Because of zero length for some internal branches, we randomly selected host species to generate host phylogenies with non-zero branch lengths and performed three replicates.”

Line 65: ERVi are endogenous; the source lineage may be in transition to colonizing the germline, for example from multiple infection events, the detection of which many may be inherently non-discoverable due to limitations in genome data. Suggest removal of “essentially” and clarification.

Response: Done as suggested! We removed “essentially”.

This point is highlighted elsewhere throughout the text. For example, line 163: “might recombine with other circulating retroviruses, generating viruses exhibiting new pathogenicity”. Recombinants with altered pathogenicities may also be formed via co-packaged ERV copies.

Response: As this reviewer pointed out, co-packaging of two complementary defective ERV copies might promote the formation of infective viral particles. We added this scenario in the Introduction and Discussion section.

Line 155: “...and elongates the list of retroviruses being circulating in the wild...”. The study does indeed add to our knowledge of retrovirus diversity, but the reviewer suggest caution on the authors’ statement that the data represent ERVs with counterparts that are “being circulated” as the data do not confirm current infectious viruses but assesses a captured endogenous copy. This point raises concern of the discrepancy of the difficulties in age estimates of the ERVi discovered and characterized by the authors. Though it is prudent to infer the most recent integrants as having identical LTRs (as well as ORFs and unfixed status), estimates of formation could be loosely inferred by LTR-LTR comparison, and because this approach is limited to the host neutral mutation rate the observations of 0 changes between LTRs is still limited to estimates at least in the hundreds of thousands of years. Having comparison of orthologous loci may narrow this range, as well as genotyping to determine a ‘degree’ of insertionally polymorphism, but these findings can not be used to confirm the presence of contemporary viruses still circulating among the surveyed populations. To clarify for the audience, please edit this statement to reflect inference of recently and/or putatively ERVi relatives in those populations.

Response: Thanks for the great and helpful comment. As this reviewer correctly pointed out, even a pair of identical LTRs may have evolved for hundreds of thousands of years. For each ERVi, we tried to identify orthologous loci within closely related species, but did not identify any orthologous loci within closely related species, indicating the recency of ERVi. On the other hand, this reviewer correctly pointed out that this cannot be used to confirm the presence of contemporary viruses that are circulating in the surveyed population. In the revised manuscript, we added some discussion for the aforementioned limitation in the Discussion section. As suggested, ERVi are not necessarily exogenous retroviruses. More precisely, ERVi can represent retroviruses that are recently or currently circulating in their host population. Therefore, we removed and tuned down related claims throughout the revised manuscript.

Minor Concerns:

Line13 suggest: “... and show that at least 412 ERVi representing 217 viral operational

taxonomic units...”

Response: Done as suggested!

The Introduction requires some streamlining and addressed concerns:

Line 26: suggest “...integrated proviruses may become vertically...”, or “have the potential to”

Response: Done as suggested!

Line 28: suggest removing “Serially documenting the past retroviral infections” and perhaps replacing with “Replicated as a permanent part of the host genome”

Response: Done as suggested!

Lines 38: please clarify “in the early phases”

Response: Done as suggested!

Line 42: suggest “... rarely documents, however examples exist such as the Koala...”

Response: Done as suggested!

Line 39: suggest removing the sentences “Meanwhile, various disruptive mutations begin to occur in ERVs”

Response: Done as suggested!

Line 43: suggest “...we performed phylogenomic mining...”

Response: Done as suggested!

Line 44: The authors state more than once “over” 2,004 genomes; please remove for specificity of analyses using the 2,004 versus additional analysis of SRA data for screening of insertional polymorphism.

Response: Done as suggested!

Line 47: suggest “...vertebrates and expands our knowledge of the diversity”

Response: Done as suggested!

Line 66: suggest removing “the”, per “circulating in the vertebrates”

Response: Done as suggested!

Line 77: suggest add reference PMID: 27001843 (Halo et al, 2019)

Response: We cited this literature (PMID: 27001843) in the original manuscript.

Line 140: please define EIDs (line 168)

Response: Done as suggested!

Line 150: suggest add reference PMID: 25535393 (Hayward et al, 2014)

Response: Done as suggested!

Reviewer #3 (Remarks to the Author):

This manuscript would seem to make a compelling case that (a) germ-line infections by retroviruses are much commoner than previously shown and (b) that the diversity of retroviruses is much greater than previously thought. These are important conclusions and I think both are correct. However, in some respects the manuscript tends to resemble an extended abstract. There is almost too much information included here and perhaps too little description of how it was put together or enough consideration given to the significance of qualitative differences between the different viruses. In addition, there seemed to be a tendency to state results rather than to show and interpret data leading to a conclusion. Moreover, it was sometimes hard to find the information needed to evaluate all the statements made. There are several areas which I would like to see addressed in more detail.

Response: Thanks for the appreciation and all the great and helpful comments! We revised the manuscript, and please see the following point-to-point response.

First, is it possible that some of the new proviruses in mice do not reflect new genomic invasions but rather the result of amplification of ERVs which are already fixed in the population and thus not considered an ERVi? The clusters of new ERVs in *M. mus* (Mmus.ERVi.1a-j in F3) and *M. spr* (Mspr. ERVi.3a-g in F3S) would seem consistent with such a view. How different are they from one another – are alignments readily available? It might be worth assembling a list of intact ERVs that do not show insertional polymorphisms

– sequence comparisons with the ERVi list might shed light on the latter’s origins.

Response: Amplification accompanies invasion, and they cannot be disentangled. At the population level, the frequency of an ERV insertion changes under selection and/or drift. And at the level of genome, ERV amplifies within the genomes. ERVi we identified in this study actually represent a continuous, rather than discrete, spectrum of invading retroviruses, covering the very beginning to the relatively late stages of retroviral invasions. Moreover, in general, if an ERV is already fixed in the host population, it is highly likely defective and cannot produce ineffective ERVi anymore.

The sequence identity among different Mmus.ERVi.1 ranges from 96.82% to 99.97%, and the sequence identity among different Mspr.ERVi3 ranges from 99.05% to 99.92%. For each of them, sequences are highly similar. All Mmus.ERVi identified in this study were insertionally polymorphic and have not yet been fixed in the mouse population. Therefore, we think Mmus.ERVi.1 and Mspr.ERVi3 are generated by retroviral invasions (and of course amplification).

Second, considering the Mus viruses it is worth pointing out that many would not consider Musculus one distinct species – are all 8 vOTUs equally distributed in domesticus/musculus/castaneus?

Response: Thanks for pointing this out. In this study, we did not distinguish *Mus musculus* subspecies, because a high proportion of individuals from which genome data are available cannot be attributed to specific subspecies. Our dataset includes 73 *M. musculus domesticus*, 16 *M. musculus musculus*, 2 *M. musculus castaneus*, 3 *M. musculus helgolandicus*, and hundreds of unclassified *M. musculus* genomic sequencing data. The 8 vOTUs exhibit different distribution in the four mentioned subspecies (see the following table). However, this pattern should be taken with caution due to the limited number of sequencing data available for *M. musculus castaneus* and *M. musculus helgolandicus*.

Subspecies	Mmus. ERVi.1	Mmus. ERVi.2	Mmus. ERVi.3	Mmus. ERVi.4	Mmus. ERVi.5	Mmus. ERVi.6	Mmus. ERVi.7	Mmus. ERVi.8
Mus musculus castaneus	0/2	0/1	-	0/2	0/1	0/2	0/2	0/1
Mus musculus domesticus	18/73	0/68	38/65	50/67	35/65	0/67	0/73	0/67
Mus musculus helgolandicus	3/3	0/3	0/3	3/3	2/3	0/3	1/3	0/3
Mus musculus musculus	1/16	1/16	0/11	1/4	1/10	1/16	1/10	0/10

Might some of the apparent invasion events during the generation of laboratory mice be explained if one or more their parents were not included in the random sample?

Response: This reviewer correctly points out that an ERVi insertion might be derived from their wild parents. We have added the possibility in the revised manuscript. Thanks for this helpful comment.

Third, I wonder whether the diversity of retroviruses is not altogether unanticipated. See Gifford et al *Retrovirology* 15:59, 2018 for a discussion of some of the difficulties in establishing a unified ERV/retrovirus phylogeny. It would be interesting to see a comparison between the tree in F2 of the current manuscript with that of F4 in the proceeding paper with its suggestion of “three ‘placeholder’ groups designed to act as temporary ‘bins’ for ERV loci that cannot be confidently placed within the existing taxonomic system approved by the ICTV”. Again, it would be nice to have easily accessible protein alignments.

Response: Excellent point! Indeed, it is difficult to establish a unified ERV/retrovirus phylogeny, and it is also difficult to establish the relationship between ERVi and the three placeholders (the actual diversity of the three placeholders are not known). Moreover, the three placeholders lack extant exogenous virus representatives. In the revised manuscript, we cited the Gifford paper, and refined (for example, clarifying that we expand the diversity of retroviruses that have been recently circulating in the wild) and turned down (for example, removing greatly) the related statements. In the revised manuscript, we also added the alignment of RT proteins used in Fig 2B as Dataset S1.

REVIEWERS' COMMENTS

Reviewer #2 (Remarks to the Author):

The authors have addressed relevant concerns by all reviewers; one concern remains for this review -- please remove multiple references to "tera-base" or "TB" data; 2004 genomes is sufficient.

Reviewer #3 (Remarks to the Author):

I think the revised manuscript improved and represents a significant resource. However, I think it essential that the DNA sequences of all the novel ERVIs be easily available for downloading without going back to the original genome sequences. Further, regarding the statements about ERVi diversity, I think it would be helpful to see where representative members of the HERV-K, -H and -W families as well as the IAPs map in the analysis illustrated in Fig 2b. Finally, a question that is perhaps unanswerable: have retroviruses ever been demonstrated to contribute to the extinction of a species? One can imagine how they might, but what is the evidence that they have?

RESPONSE TO REVIEWERS' COMMENTS

Reviewer #2:

The authors have addressed relevant concerns by all reviewers; one concern remains for this review -- please remove multiple references to "tera-base" or "TB" data; 2004 genomes is sufficient.

Response: Thanks! As suggested, we removed “tera-base scale” in the revised manuscript.

Reviewer #3:

I think the revised manuscript improved and represents a significant resource. However, I think it essential that the DNA sequences of all the novel ERVIs be easily available for downloading without going back to the original genome sequences. Further, regarding the statements about ERVi diversity, I think it would be helpful to see where representative members of the HERV-K, -H and -W families as well as the IAPs map in the analysis illustrated in Fig 2b. Finally, a question that is perhaps unanswerable: have retroviruses ever been demonstrated to contribute to the extinction of a species? One can imagine how they might, but what is the evidence that they have?

Response: Thanks! We deposited all the DNA sequences of all the novel ERVi to figshare (doi: 10.6084/m9.figshare.23653254). HERV-K, -H, and -W as well as IAPs were not used in the analysis illustrated in Fig 2b. To our knowledge, no retrovirus has been demonstrated to contribute to the extinction of a species.